# The Landscape of Pediatric Precision Oncology: Program Design, Actionable Alterations, and Clinical Trial Development

**DOI:** 10.3390/cancers13174324

**Published:** 2021-08-27

**Authors:** Karin P. S. Langenberg, Eleonora J. Looze, Jan J. Molenaar

**Affiliations:** 1Princess Máxima Center for Pediatric Oncology, Heidelberglaan 25, 3584 CS Utrecht, The Netherlands; e.j.looze@amsterdamumc.nl (E.J.L.); j.j.molenaar@prinsesmaximacentrum.nl (J.J.M.); 2Department of Pharmaceutical Sciences, Utrecht University, P.O. Box 80082, 3508 TB Utrecht, The Netherlands

**Keywords:** precision medicine, targeted therapy, next-generation sequencing

## Abstract

**Simple Summary:**

Precision medicine is a revolutionary new way to deliver cancer treatment by targeting specific genetic changes of the cancer of the individual child with the goal of improving cure rates and reducing toxicity. In this review, we illustrate the evolution of cancer treatment in this groundbreaking new era. We compare characteristics and early results of precision medicine programs in pediatric oncology as well as novel clinical trial initiatives translating these findings into potential clinical benefit for all children and adolescents with cancer.

**Abstract:**

Over the last years, various precision medicine programs have been developed for pediatric patients with high-risk, relapsed, or refractory malignancies, selecting patients for targeted treatment through comprehensive molecular profiling. In this review, we describe characteristics of these initiatives, demonstrating the feasibility and potential of molecular-driven precision medicine. Actionable events are identified in a significant subset of patients, although comparing results is complicated due to the lack of a standardized definition of actionable alterations and the different molecular profiling strategies used. The first biomarker-driven trials for childhood cancer have been initiated, but until now the effect of precision medicine on clinical outcome has only been reported for a small number of patients, demonstrating clinical benefit in some. Future perspectives include the incorporation of novel approaches such as liquid biopsies and immune monitoring as well as innovative collaborative trial design including combination strategies, and the development of agents specifically targeting aberrations in childhood malignancies.

## 1. Introduction

Cancer remains the leading cause of death in children and adolescents in high-income countries: about one in five children with cancer will succumb to their disease [1,2]. Despite major advances through intensification of cytotoxic chemotherapy, optimizing local treatment and perfection of supportive care, prognosis for high-risk and refractory cancers remains poor, especially for metastatic sarcoma, high-risk neuroblastoma, several types of central nervous system (CNS) tumors, acute myeloid leukemia (AML) and rare pediatric cancers [3,4]. Moreover, improved survival has come at a high cost: survivors are facing serious late side effects of intense multimodality therapy, including infertility, cardiomyopathy, neurocognitive sequelae as well as secondary malignancies, significantly impacting quality of life [5]. Therefore, it is imperative to develop more effective and less toxic treatments for all 400,000 children and adolescents of 0–19 years diagnosed with cancer globally each year [1].

In this review, we summarize the evolution of genomics-informed precision medicine, focused on the results of established pediatric oncology programs worldwide, discussing challenges and opportunities to accelerate the implementation of pediatric precision oncology.

### Methods

To identify pediatric precision medicine studies, a PubMed search was performed with the following search term [”childhood cancers” OR “pediatric oncology” AND “precision medicine” OR “targeted therapy”] up until 1 December 2020. All found studies were uploaded in Rayyan (http://rayyan.qcri.org; accessed on 2 December 2020) and subsequently assessed on both title and abstract. After inclusion, the reference lists of respective studies and pediatric precision medicine reviews were searched for additional relevant pediatric precision medicine studies. Abstracts and unpublished data were also considered. Studies focused on adult precision medicine that did not separately report results for children and adolescents were excluded. For the independent precision medicine programs, the following data were collected: program name, country, study period, number of patients included, number of patients analyzed, cancer types included, maximum age for inclusion, inclusion criteria (primary high-risk, relapse or refractory), types of molecular analysis performed, percentage of (actionable) somatic and germline events, change in diagnosis or re-classification, target priority score, percentage of targeted therapy applied, and clinical outcome. Before performing the analysis, collected data were sent to their respective authors/project leaders for verification. Data were analyzed using descriptive statistics.

## 2. Molecularly Informed Personalized Medicine in Adult Oncology

It is hypothesized that matching treatments to molecular changes in the tumor results in more effective cancer control and less long-term treatment-related side effects [6]. Rapid evolution in sequencing technologies and computational analyses have identified cancer drivers and druggable molecular alterations, changing the paradigm in oncology from histology-based diagnostics and subsequent cytotoxic treatment to using whole-genome, whole-exome, whole-transcriptome, and/or whole-methylome data to select the optimal treatment for individual patients. The pharmaceutical industry prioritized developing novel agents that target genes commonly mutated in adult malignancies. Important clinical progress was achieved in adults with *BCR-ABL* fusion positive chronic myeloid leukemia [7], *HER2*-positive breast cancer [8], lung cancer harboring *EGFR* mutations [9] or *EML4-ALK* translocations [10] as well as *BRAF-V600E* mutated melanoma with BRAF and MEK inhibitors [11].

Several adult precision medicine trials have been initiated, characterizing genomic alterations to select a targeted therapy, such as Bisgrove [12], IMPACT [13,14,15], MOSCATO [16], NCI-MATCH [17,18] and NCI-MPACT [19,20], PREDICT [21,22,23], MyPathway [24], ProfiLER [25] WINTHER [26], and the Drug Rediscovery protocol [27], as reviewed by Fountzilas [28], Tsimberidou [29] and Gambardella [30]. Despite early signals of activity, clinical benefit of personalized treatments has only been identified in some specific subgroups. In the SHIVA trial, no improvement of progression-free survival was observed when off-label use of molecularly targeted agents was compared with standard treatment [31].

## 3. The Differing Genomic Landscapes of Childhood and Adult Cancers

Pediatric pan-cancer genome and transcriptome studies reveal a landscape that differs substantially from adult malignancies [32,33,34,35]. Childhood malignancies commonly occur in developing mesodermic rather than adult epithelial tissues. Whereas many adult tumors are characterized by a high number of somatic mutations, pediatric cancers typically have few [36]. Mutation rates vary across cancer types, ranging from 0.02 mutations per megabase in hepatoblastoma to 0.49 in Burkitt’s lymphoma and correlate significantly with age. Hypermutator phenotypes are uncommon, except in children who carry mutations in genes that code for DNA damage repair mechanisms [33,37]. Pediatric cancers are frequently defined by a single driver gene as opposed to the multiple cancer-driving mutations identified in adults [32,33]. Driver mutations are more exclusive in specific tumor types, whilst adult cancers more frequently share mutations [33]. Furthermore, childhood tumors harbor a unique spectrum of mutations: only 30–45% of cancer driver genes overlap with adult pan-cancer analyses [33]. The most commonly mutated set of genes are those involved in epigenetic modification (25% of tumors) [33]. Over 50% of all tumors harbor potentially druggable mutations, most commonly in the MAPK, cell-cycle, or DNA-repair pathways [33].

Distinct mutational signatures are identified by whole-genome sequencing [32,33]. Structural variations and copy number alterations play an important role in over 60% of pediatric cancers, stressing the need for not only mutation evaluation but complementary approaches to detect clinically relevant molecular events [32,33,34,38] (Figure 1). In 7.6% of cases, tumors were associated with predisposing germline mutations, mostly in DNA repair genes, potentially creating opportunities for immunotherapy in a subset of these patients [39]. These large-scale analyses of childhood tumor genomes led not only to substantial insights into cancer development, but also identified potentially druggable events, setting the stage for the introduction of precision medicine programs in pediatric oncology [32,33].

## 4. The Development of Precision Medicine Programs in Pediatric Oncology

Over the last decade, several large-scale national pediatric precision oncology programs have been initiated, enrolling over 3000 children, adolescents, and young adults (AYA), as published up until December first of 2020 (Figure 2). These studies investigated the potential of molecular-driven precision medicine and began to assess the clinical benefit of targeted therapies.

The following precision medicine programs were developed: the United States initiated BASIC3 [40], MSKCC PMTB [41], PIPseq [42,43], Peds-MiOncoSeq [44], ClinOmics [45], UCSF [46], iCAT [47] and pediatric MATCH [48]; Canada initiated PROFYLE [49], TRICEPS [50] and KiCS [51]; France initiated MMB [52], MOSCATO-01 [53], ProfiLER [54] and MAPPYACTS [55]; Australia initiated TARGET [56] and the Zero Childhood Cancer Program [38]; Germany initiated the INFORM study [39,57]; the Netherlands initiated the individualized THERapy (iTHER) program [58,59]; the United Kingdom initiated SMPaeds [60]; Korea initiated SMC [61] and finally the transnationalPacific Pediatric Neuro-oncology consortium was initiated [62]. An overview of reviewed precision medicine program characteristics is shown in Table 1 and Figure 2.

Internationally published and/or presented results demonstrate the feasibility and opportunities of molecular-driven precision medicine and revealing a rate of actionable variants that justify the development of predictive biomarker-driven trials for childhood cancer. In addition to the detection of potentially druggable events, molecular profiling could also be used to identify germline mutations and change or refine diagnosis [63,64,65,66,67,68,69,70]. We will discuss these aspects separately in the next sections.

## 5. Patient Accrual

Enrollment criteria for patients to enter precision medicine programs are heterogeneous. The majority of precision medicine programs include children, adolescent, and young adults with various tumor subtypes at different time points, although the published cohorts consisted mainly of patients with refractory or relapsed CNS malignancies as well as extracranial solid tumors. Several initiatives aim to inform physicians of clinically actionable targets to promote enrollment in pediatric early clinical phase trials to investigate whether it is effective to treat cancer in children and adolescents by targeting certain genetic changes in their tumors with specific targeted drugs. Other programs aim to increase our understanding of genetic drivers of pediatric cancer and to identify new clinically relevant subtypes.

Although most programs included relapsed and/or refractory patients, a debate is ongoing on the timing of inclusion of patients. Several distinct points can be highlighted. First, at diagnosis, patients with standard-risk disease might not benefit from identifying additional treatment options, but refinement of diagnosis as well as disease classification can be crucial for subtypes in CNS tumors as well as sarcoma or tumors of unknown origin [71,72]. Second, high-risk tumors may show single pathway addiction at diagnosis and might respond better to targeted inhibition when incorporated early into treatment regimens [73,74]. Nearly 50% of primary childhood tumors harbor a potentially targetable genetic event [33], and treatment strategies are already being adapted for certain subtypes. In Philadelphia chromosome-positive acute lymphoblastic leukemia, the introduction of increasingly potent tyrosine kinas inhibitors (TKIs) has revolutionized therapy [75]. In neuro-oncology, single-agent dabrafenib in pediatric patients with *BRAF* V600–mutant relapsed and those with refractory low-grade glioma showed a 44% objective response rate [76]. In non-CNS solid tumor patients aged 1 month to 21 years whose tumors harbor an *NTRK* fusion gene, larotrectinib, a selective TRK inhibitor, had a 93% objective response rate [73,74]. In children with newly diagnosed high-risk neuroblastoma, clinical trials are ongoing in North America and Europe targeting *ALK* aberrations by adding ALK-inhibitors to first-line therapy, with molecular profiling transitioning from basic science to validation in the clinic (NCT03126916; NCT04221035). Finally, there is a subset of patients who do not achieve meaningful responses after induction and therefore harbor a poor prognosis with conventional treatment protocols. Thus, identifying actionable events early in the disease course might provide unique treatment options in a subset of very-high-risk patients.

Druggable events vary between primary and relapse tumors. Only 37% of primary tumors retained these events upon progression whilst most tumors gained events, as reported in disease-specific reports [32,33,77,78,79,80,81,82,83,84] These substantial spatiotemporal differences in the molecular profiles of multiple samples acquired from the same patient as well as metastases compared to primary tumors indicate the need for subsequent analyses to optimize biomarker-driven selection in clinical trial recruitment [39].

## 6. Next-Generation Sequencing, Data Integration and Visualization

Several prospective precision medicine program initiatives have shown feasibility of integrating genomic and epigenomic data in real time to direct treatment decisions for pediatric patients. Applied methods to sequence somatic as well as germline DNA vary, as summarized in Table 1: from targeted cancer gene panel sequencing; whole exome sequencing (WES—with or without computational analyses focusing on a predefined gene list of known cancer genes); to sequencing of the full genome (whole-genome sequencing, WGS). RNA sequencing (RNAseq) and RNA microarrays can be used to detect actionable fusion genes and analyze expression patterns for target identification as well as subgroup refinement. Methylome profiling is incorporated in some programs to classify tumors as well assess the methylation status of relevant genes. Each molecular profiling platform analyzed their data centrally via dedicated bioinformatics pipelines to predict pathogenic variants.

Data sharing thus far is limited and can be challenging due to privacy regulations, but several platforms aim to analyze and publicly visualize genomic data since effective data sharing is key to accelerating research. For example, St. Jude Cloud is an expanding cloud-based data-sharing ecosystem with genomic data from >10,000 pediatric patients (https://www.stjude.cloud; accessed on 11 May 2021) [85]. Another web-based genomics analysis and visualization application extensively used by the pediatric community including the “Innovative Therapies for Children with Cancer Paediatric Preclinical Proof-of-concept Platform” (ITCC-P4) is the R2 Genomics Analysis and Visualization Platform, which integrates genomic and clinical data as well as in vitro and in vivo model systems and drug sensitivity profiles (http://r2.amc.nl; accessed on 11 May 2021) [86].

To date, no comparisons between precision medicine platforms and strategies have been published and health technology assessment is lacking. Whilst the most optimal approach is still unclear and the optimal molecular profiling approach might vary by disease type and stage, several technologies transition to become the standard of care in developed countries. Institutions will choose NGS approaches based on the quality and quantity of available material, clinical relevance, and research interests as well as sustainable funding opportunities.

## 7. Translating Molecular Findings into Clinic: Identification and Prioritization of Targets

After sequencing and bioinformatic processing of raw data, molecular data are integrated with (pre)clinical evidence to select clinically relevant alterations. In all programs, an expert review is performed by a molecular tumor board, comprising experts of various disciplines such as molecular biologists, pediatric oncologists, clinical geneticists, pathologists and/or early-phase clinical trial physicians and pharmacists.

The percentage of actionable alterations that were reported in the different studies range from 27% to 100% (Table 1). However, there is no standardization on what is considered actionable, and therefore numbers should be interpreted with caution. For instance, pediatric MATCH defined an alteration as being actionable only when there is a treatment arm available in a phase II clinical study. Consequently, actionability relies on the availability of a targeted agent and was therefore variable over time. Moreover, the “druggability” of any event will be impacted as we gain more insight from preclinical studies and novel drug development. Some precision medicine studies, including TRICEPS, defined non-druggable alterations as being actionable if they informed diagnosis, prognosis, or treatment stratification.

Additionally, prioritization of the detected events differs between precision medicine programs. A common classification system that is used for the recommendation of targeted therapies is the NCI-MATCH tier. Actionable alterations are ranked from high to low level of supporting evidence: Tier 1, clinical evidence in the same cancer; Tier 2, clinical evidence in a different cancer; Tier 3, preclinical evidence in the same cancer; Tier 4, preclinical evidence in a different cancer type [45]. The INFORM consortium on the other hand developed a 7-scale target prioritization algorithm, taking into account the type of alteration, the mechanism of action of potential drugs within the pathway, the level of evidence for the specific alteration, and its role in the specific cancer type [39].

In conclusion, interpretation and prioritization of actionable event calling is challenging and dynamic in a rapidly evolving landscape of new biomarkers and treatments. Further optimization and standardization of the process of target prioritization will be crucial to allow for the comparison of molecular profiling technologies as well as guide clinical decision making.

## 8. Germline Variants

Most precision medicine studies highlight the importance of germline alterations in cancer-related genes, and cancer predisposition syndromes are recognized as an important cause of pediatric cancer development [87].

Detecting variants differs between programs due to distinct inclusion criteria and various gene panels. The mean percentage of germline mutations detected by molecular profiling is 14% with a minimum of 6% and a maximum of 35% (Table 1). Variants are usually classified into five categories according to American College of Medical Genetics and Genomics (ACMG) guidelines: pathogenic (class 5), likely pathogenic (class 4), variant of unknown significance (class 3), likely benign (class 2), and benign (class 1) [88]. The high percentage of germline findings shows the relevance of genomic analysis on combined tumor and germline DNA.

Germline pathogenic variants can be linked to somatic features of the tumors, identifying potential treatment strategies. High tumor mutational burden is detected in patients with constitutive mismatch repair deficiency based on biallelic germline loss of MSH6 or PMS2. Enrichment in mutational signature 3 (‘BRCAness’) can be found in tumors from patients with germline homologous recombination defects [37,51].

Therefore, molecular profiling not only has the potential to confirm a mutation in a cancer predisposition gene, but also to guide treatment in cases where germline alterations were not predicted by family history or not clinically evident [89]. Moreover, patient and family members could be referred for genetic counselling and cancer surveillance, possibly contributing to early tumor detection associated with improved long-term survival [90].

There are no structured reports of pharmacogenomic alterations and research on clinical impact of variants on pharmacokinetics, and pharmacodynamics in pediatric oncology is still in its infancy [91].

## 9. Change or Refinement of Diagnosis

Next to the identification of somatic alterations that can be targeted by a specific treatment, molecular profiling also has the potential to lead to a clinically relevant change or refinement in diagnosis, for example, through the identification of a specific fusion or DNA methylation-based classification.

Tumor re-classification or changing or refinement of diagnosis was possible for a substantial number of patients (Table 1). These results support molecular profiling at an early stage as it informs treatment strategy. Classification of brain tumors or sarcoma based on methylome is being advanced into clinical care [71,72].

## 10. Targeted Therapy and Clinical Decision Making

Molecular tumor boards (MTBs) provide an individual report, summarizing all actionable genomic aberrations and matched treatment and/or clinical trial recommendations. The aim is to help clinicians to translate molecular profiles into clinical benefit, maximizing the impact of precision medicine. Optimal design of these MTBs has not been determined [92].

The time to results varied considerably between the precision medicine programs (Table 1), depending upon the entry criteria and the NGS techniques and computational pipelines used. Turnaround time can be relevant, especially in a relapse setting, since performance status is an important inclusion criterium for phase I/II clinical trials [52].

The decision to apply targeted therapy based on the MTB recommendation is made by the treating pediatric oncologist together with the patient and parents. Critical decision-making factors in this process remain to be elucidated. However, patient performance status might be a limiting factor since children are often enrolled with end-stage disease and subsequently deteriorate or die early. In addition, turnaround time between biopsy and molecular tumor board results requires initiating an alternative conventional (palliative) treatment protocol, balancing toxicity, and quality of life. Additionally, there might be difficulties accessing targeted therapy drugs, particularly in children. Since only few molecular targeted drugs have pediatric indications, a targeted therapy could either be received through enrolment in a phase I/II clinical trial, by off-label or compassionate approaches [93]. On average, 27% of the patients (rates ranging from 3% to 58%, Table 1) received targeted therapy based on the recommendation of the MTB. The ratio between the percentage of actionable alterations and number of patients that received targeted therapy differed considerably between precision medicine programs. This could be explained by profound differences in trial design and follow-up time, the varying molecular profiling strategies used, the lack of standardization of actionable event identification and prioritization, as well as regulatory and logistical challenges in obtaining the matched targeted drug. Therefore, we recommend interpreting these results with caution.

## 11. Clinical Benefit

For 5 out of the 18 included precision medicine programs in this study, follow-up after the identification of actionable alterations is not (yet) published, and others reveal contradictory results in a non-randomized setting. Encouraging results have recently been published from large-scale studies in Europe and Australia. Collaborative data from INFORM and iTHER showed increased progression-free survival for the subgroup of patients that followed treatment recommendation for a very-high-priority target [57]. The Zero Childhood Cancer Program demonstrated that the clinical outcome of the patients treated with a targeted agent was favorable compared to patients included in unselected phase I clinical trials. Remarkably, clinical outcome did not correlate with the tier score of the recommended targeted agent [38]. Previously published results of the Peds-MiOncoSeq study mentioned that 9/15 patients showed partial response and one complete remission [44]. This is opposed to a lack of clinical response in the iCAT study [47]. Similarly, the Pacific Pediatric Neuro-oncology consortium showed comparable median overall survival between targeted and cytotoxic therapy [62]. These conflicting results, again, should be interpreted with caution, as studies, methods and patients are heterogeneous, stressing the need for harmonization and collaboration.

## 12. Clinical Trial Development: Innovative Global Collaboration

Currently, the availability of approved molecular targeted drugs for pediatric patients is still limited compared to adult indications and many new targeted drugs lack dosage guidelines and efficacy data in children [93]. Targeted therapy development is complicated by the fact that pediatric malignancies show a relative paucity of targetable mutations as well as distinct molecular alterations compared to adult cancers, suggesting that new therapeutic agents are required for pediatric cancer. In addition, there is a lack of available clinical trials and a smaller number of eligible patients for each study.

Innovative strategies in early drug development for children, adolescents and young adults have been proposed by several collaborative groups [64,67,94,95,96,97]. For example, the pediatric platform ACCELERATE, comprising multiple stakeholders in pediatric oncology, is aiming for biology-driven early drug development and clinical trial design for children and adolescents with cancer [98]. In addition, recent regulatory measures, such as the Research to Accelerate Cure and Equity for Children Act (RACE Act), are attempting to stimulate earlier access to novel agents for children and adolescents with cancer [96].

Recently, large-scale pediatric trial initiatives have been developed in order to design phase I/II (combination) trials. Basket trials are designed to enroll biomarker-selected patients with many different cancer types who are assigned to one of the biology-matched subprotocols [97]. Examples of ongoing pediatric basket trials are AcSé-ESMART (NCT02813135) [53,99], INFORM2 (NCT03838042) [100], and Pediatric MATCH (NCT03155620) [101].

## 13. Ongoing and Future Perspectives in Pediatric Precision Oncology

### 13.1. Patient-Derived Models and Drug Sensitivity Profiling

In addition to state-of-the-art molecular profiling, several precision medicine programs are adding functional testing of drug sensitivities in patient-derived models to complement current genomic approaches.

Patient-derived xenograft (PDX) models generated from the transplantation of patient tumor cells into immunodeficient mice or zebrafish conserve the original tumor characteristics preserving the heterogeneity [102,103]. Functional drug testing in vivo has been incorporated into the ZERO Childhood Cancer program and is explored in several others, although time to engraftment, costs, as well as ethical considerations remain challenging. The development and molecular characterization of large numbers of pediatric cancer PDX models has been undertaken both in Europe (ITCC-P4) and the US PPTP/C [104].

Patient-derived organoids resemble in vivo tumors, model treatment response and hold promise to predict drug response in a personalized fashion. They are established with a high success rate and are readily available for drug sensitivity testing [105,106,107,108,109,110,111]. INFORM, MAPPYACTS, iTHER and Zero are collaborating in the COMPASS consortium (ERAPERMED2018-121; Clinical implementation of Multidimensional Phenotypical drug Sensitivities in pediatric precision oncology—ERA-LEARN) to establish a standardized ex vivo drug sensitivity testing platform and to evaluate the incorporation of direct functional testing for efficacies of cancer drugs for individual patients.

### 13.2. Emerging Technologies: Liquid Biopsies

Several reports have demonstrated the feasibility of detecting tumor DNA in liquid biopsies using NGS or droplet digital polymerase chain reaction, including for pediatric cancers [112,113,114]. Evaluation of tumor heterogeneity and clonal selection due to treatment pressure is adequately reflected in samples and might be a non-invasive alternative to repetitive multifocal biopsies, contributing to patient monitoring and personalized treatment. The impact of the surrogate approach based on cfDNA testing to identify targetable genetic alterations is currently under evaluation in several pediatric precision oncology programs, including MAPPYACTS [55] and SMPaeds [60].

### 13.3. Novel Therapies: Immune Interventions

Immunotherapy has been an exciting new development in systemic cancer treatment in a range of adult cancers such as melanoma and lung cancer. In pediatrics, response rates and outcomes significantly improved in patients with relapsed and refractory hematological malignancies such as B-ALL and lymphoma with antibody-based therapy including blinatumomab and inotuzumab ozogamicin as well as CAR-T therapy [115]. Anti-GD2 therapy with dinutuximab increased the survival of patients with high-risk neuroblastoma and is implemented into frontline therapy [116]. Immune checkpoint inhibition holds great promise in children with an inherited deficiency in DNA mismatch repair [117], and many clinical trials are ongoing to explore opportunities in childhood cancer [118]. However, one of the defining traits of pediatric tumors is their low mutational burden and relative lack of neoantigen expression, which limits their susceptibility to immune targeting. In addition, many immunotherapies lack reliable predictive biomarkers. Consequently, precision medicine programs focus on ancillary studies and novel techniques such as high-dimensional characterization of the immune infiltrate with the goal to increase the number of patients who can be linked to an effective immunotherapeutic regimen in the future.

### 13.4. Clinical Trials: Incorporating Combination Strategies

Precision medicine trials with single-agent small molecules have shown limited success. Studies indicate heterogeneity of molecular mechanisms that can drive tumorigenesis within one tumor type, making it unlikely to improve curation rates with a new single treatment modality [68,119,120]. Moreover, it may be challenging to differentiate driver from passenger molecular alterations, and additional pharmacogenomic, pharmacodynamic or kinetic aspects should be researched as well [30].

Future trial designs will include biologically driven combinations of molecularly targeted therapies as well as targeted treatments combined with chemotherapy or immunotherapy, as initiated by AcSé-ESMART (NCT02813135) [54,99] as well as INFORM2 (NCT03838042) [100]. As an example, AcSé-ESMART Arm G assessed the activity and safety of nivolumab in combination with metronomic cyclophosphamide with or without irradiation, as per the physician’s choice. The primary endpoint was objective response rate. Thirteen patients were treated. Nivolumab in combination with cyclophosphamide was well tolerated but had limited activity, and metronomic cyclophosphamide did not modulate systemic immune response [99].

In addition to innovative trial design, global collaboration facilitated by sustainable funding is necessary to identify and enroll eligible patients for each study [97,121]. A prime example includes the phase 3 clinical trial developed by the Children’s Oncology Group (COG) and the International Society of Paediatric Oncology Europe Neuroblastoma (SIOPEN), funded by Solving Kids’ Cancer UK and six partner charities. The study that is scheduled for 2021 and known as TITAN—Transatlantic Integration Targeting ALK in Neuroblastoma—is a promising example of collaboration between these North American and European neuroblastoma consortia.

Despite global collaboration and regulatory changes, several challenges remain. As molecular enrichment in a trial arm does not always take place, it is difficult to determine the clinical outcome for the biomarker-selected patients. Moreover, as basket trials do not have a control group, the potential to assess whether clinical outcome can be improved by molecular selected targeted therapy compared to conventional treatment is limited. Another challenge is caused by tumor complexity and resistance, which makes it unlikely that targeted monotherapy would result in complete remission. More extensive preclinical research is needed to identify the genomic alteration—drug combinations that could be effective. Therefore, in the future, international coordination will be crucial to generate a database to inform rational trial design and to evaluate combination trials, paired with conventional or combined targeted therapy, in enriched cohorts. In addition, little is known about the long-term toxicities of most novel targeted and immunotherapy agents. To address this gap, ACCELERATE has initiated the development of an international long-term follow-up prospective data registry with the aims of supporting the regulatory requirements, labeling information, and providing insight to help guide physicians and families on the appropriateness of a targeted or immune therapy for their child and inform survivorship planning [122].

### 13.5. Big Data

New methods dedicated to improving data collection, storage, processing, and interpretation continue to be developed. The collection of integrated clinical and molecular results from precision medicine initiatives as well as early phase clinical trials raises a number of challenges with respect to privacy and ethical concerns that need to be addressed to optimize progress in childhood cancer precision oncology [123,124].

## 14. Conclusions

Precision medicine in pediatric oncology has rapidly developed over the last decade. Assessing clinical benefit as well as cost-effectiveness remains challenging due to heterogeneity in patient selection as well as the lack of standardization in data interpretation and treatment recommendations. The development of innovative precision medicine trials incorporating functional model systems and novel techniques is critical to optimizing outcome. Due to global collaborative initiatives, the integration of genomic and (pre)clinical data can be used to direct the development of novel targeted agents more effectively in the future.

## Figures and Tables

**Figure 1 cancers-13-04324-f001:**
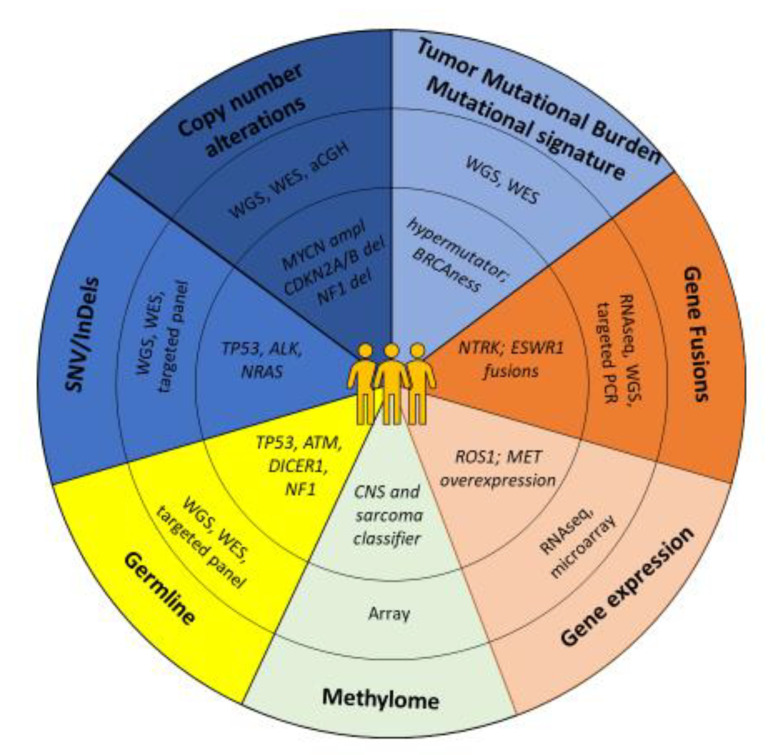
Precision medicine: techniques used with examples in pediatric oncology.

**Figure 2 cancers-13-04324-f002:**
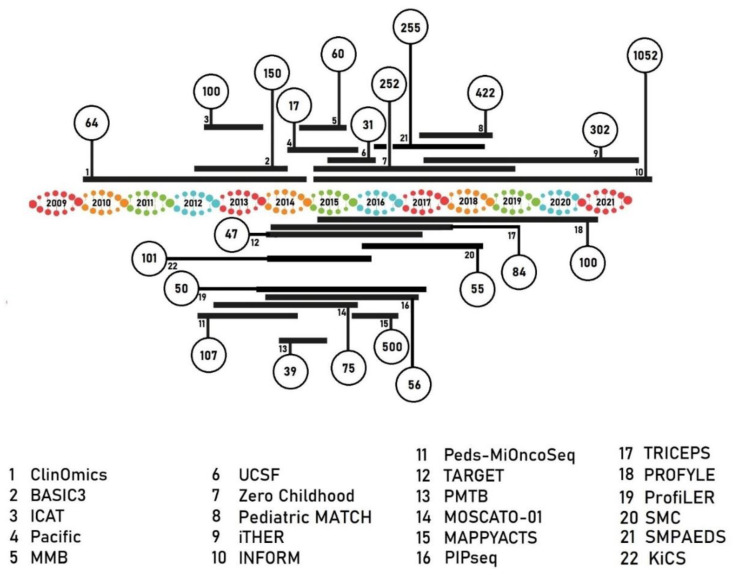
Precision medicine programs in pediatric oncology, showing the number of samples included. Horizontal bars indicating the timeframe in which patients were included.

**Table 1 cancers-13-04324-t001:** Characteristics and results of published pediatric precision medicine approaches.

Program Name/Sponsor	No. of Samples Included	No of Samples Analyzed	Inclusion Criteria	NGS Technique	Tumor Subtypes	Data Reported	Time to Results (Days)	% Actionable Alterations	% Patients Receiving Targeted Therapy (of All Samples Sequenced Successfully)	% Change or Refinement of Diagnosis	% Germline Aberrations
ClinOmics [45] *USA, NCI Center for Cancer Research*	64	59	Relapse/refractory	WES; RNAseq; SNP array	Solid tumors	Somatic & germline	NR	51	NR	7	12
Peds-MiOncoSeq [44] *USA, University of Michigan*	107	101	Primary high-risk; relapse/refractory; rare cancers	WES; RNAseq	Solid tumors; hematological malignancies	Somatic & germline	54 (average)	46	15	2	10
BASIC3 [40] *USA, Baylor College of Medicine*	150	121	Primary high-risk (newly diagnosed and untreated)	WES	Solid & CNS tumors	Somatic & germline	NR	27	NR	NR	10
iCAT [47] *USA, Dana Farber Cancer Institute*	100	89	Primary high-risk; relapse/refractory	NGS panel; aCGH	Solid tumors	Somatic & germline	NR	39	3	3	12
MOSCATO-01 [53] ** *France, Gustave Roussy Cancer Center*	75	69	Relapse/refractory	WES; NGS panel; RNAseq; aCGH	Solid & CNS tumors	Somatic & germline	19–41 (26 average)	61	19	4	10
ProfiLER [54] *France, Centre Léon Bérard*	50	43	Primary high-risk; relapse	69 gene panel; aCGH	Solid & CNS tumors; hematological malignancies	Somatic	NR	23	9	NR	NA
PIPseq [42,43] *USA, Columbia University*	-	56	Relapse/refractory; unusual presentation for age; rare cancers	WES; NGS panel; RNAseq	Hematological malignancies	Somatic & germline	40 (median)	80	13	11	24
TRICEPS [50] *** *Canada, CHU Sainte-Justine*	84	62	Relapse/refractory	WES or NGS panel; RNAseq	Solid & CNS tumors; hematological malignancies	Somatic & germline	32–120 (61 median)	87	41	22	13
PMTB [41] *USA, Memorial Sloan Kettering Cancer Center*	-	39	Primary high-risk; relapse/refractory; remission	WES; Hybrid-capture based DNA and RNA sequencing assay; RNAseq; FISH	Solid & CNS tumors; hematological malignancies	Somatic & germline	NR	73	54	NR	NR
PNOC003 [62] *Transnational, Pacific Pediatric Neuro-oncology consortium*	17	17	Primary high-risk	WES; WGS (60x); RNAseq;	CNS tumors	Somatic & germline	6–22 (13 median)	100	47	NR	NR
MMB [52] *France, Institut Curie*	60	58	Primary high-risk; relapse/refractory	NGS panel; aCGH	Solid & CNS tumors	Somatic	26–58 (42 median)	40	10	NR	NA
INFORM [39,57] *Germany, German Cancer Research Center*	1052	928	Primary high-risk; relapse/refractory	WES; lcWGS; RNAseq; 850K methylation	Solid & CNS tumors; hematological malignancies	Somatic & germline	25 (average)	85	28	7	8
TARGET [56] * *Australia, Manchester Cancer Research Centre*	-	47	Primary high-risk	NGS panel; RNAseq	Solid & CNS tumors; hematological malignancies	Somatic & germline	NR	61	NR	NR	NR
Zero Childhood Cancer [38] * *Australia, CCI*	252	252	Primary high-risk; relapse/refractory	WGS, RNAseq, 850K methylation	Solid & CNS tumors; hematological malignancies	Somatic & germline	53 (average)	71	17	5	16
PROFYLE [49] *** *Canada, The Terry Fox Research Institute*	-	100	Relapse; hard-to-treat cancer	NGS panel; WGS; RNAseq;	Solid & CNS tumors; hematological malignancies	Somatic & germline	NR	82	58	NR	14
UCSF [46] *USA, UCSF Medical Center*	31	31	Relapse/refractory; no standard therapy available	NGS panel	CNS tumors	Somatic & germline	14–21	61	NR	19	35
MAPPYACTS [55] ** *France, Gustave Roussy Cancer Center*	500	390	Relapse/refractory	WES; RNAseq	Solid & CNS tumors; hematological malignancies	Somatic & germline	NR	70	28	NR	6
SMC [61] *Republic of Korea, Samsung Medical Center*	55	53	Relapse/refractory	381 gene panel; 22 intron panel	Solid tumors	Somatic	NR	36	2	NR	NA
SMPAEDS [60] *UK, Royal Marsden Hospital*	255	209	Relapse/refractory	78 or 91 gene panel	Solid tumors	Somatic	NR	51	2	NR	NA
iTHER [58,59] *The Netherlands, Princess Máxima Center*	302	226	Primary high-risk; relapse/refractory cancers	WES; lcWGS; RNAseq; 850K methylation	Solid & CNS tumors; hematological malignancies	Somatic & germline	35 (average)	89	12	4	10
Pediatric MATCH [48] *USA, National Cancer Institute–Children’s Oncology Group*	422	357	Relapse/refractory	NGS gene panel; IHC	Solid & CNS tumors; hematological malignancies	Somatic	15 (average)	29	24	NR	NA
KiCS [51] *** *Canada, The Hospital for Sick Children (SickKids)*	-	200	Poor prognosis; rare tumors; cancer predisposition	864 gene panel; RNAseq; WGS	Solid & CNS tumors; hematological malignancies	Somatic & germline	NR	53	NR	NR	12

Data include, if known, the name and location of the program, the total number of samples included, the number of samples analyzed successfully, criteria for patient accrual, techniques used as well as type and turnaround time of the data reported, percentage of actionable events identified and percentage of patients ultimately receiving targeted therapy of all samples sequenced successfully; changed or refined diagnosis and percentage of germline alterations detected. *, **, *** precision medicine programs that are related. Abbreviations: NGS—Next-Generation Sequencing. WGS—Whole-Genome Sequencing. lcWGS—low-coverage Whole-Genome Sequencing. WES—Whole-Exome Sequencing. RNAseq—RNA sequencing. SNP array—Single Nucleotide Polymorphism array. aCGH array—Comparative Genomic Hybridization. FISH—Fluorescence in situ hybridization. IHC—Immunohistochemistry. NR—Not Reported. NA—Not Applicable.

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
