# Peer review of "The Landscape of Pediatric Precision Oncology: Program Design, Actionable Alterations, and Clinical Trial Development"

_cancers, 2021, doi:10.3390/cancers13174324_

Round 1

Reviewer 1 Report

Karin Langenberg, Eleonora Looze and Jan Molenaar present a comprehensive review of the current landscape of precision medicine in pediatric cancer. They address several key elements. The manuscript is well written and it is also very informative. The authors have to be congratulated for this effort.

I have nevertheless some comments.

Major:                   

  • The authors need to explain what has been the methodology to identify and describe the precision medicine studies that they describe in section 4:
    1. Not all the studies in Figure 1 appear in Table 1 or in the text (lines 105-114). For instance, the PRISMA study only appears in Figure 1 but not in the other two parts. Please make sure there is a consistent list of studies in the three parts (text, figure and table). If not all studies have to appear in all three parts, then please explain the reasoning.
    2. Please, also explain how the size of the circumference has been estimated: Is it by samples included/samples analyzed/patients included? If it is patients included, please then add a column with patients included in Table 1 besides the columns of samples included/samples analyzed.
    3. Please, describe what has been the search methodology to get to the published reports (e.g. timeframe of research [from 20xx to 20xx, date of last research performed, consort diagram, keywords, limits, etc; these are some examples, not all needed to be given but at least to be considered to be included). I particularly miss the pediatric study PROFILE (https://ascopubs.org/doi/full/10.1200/PO.20.00023); also, not clear to me what has been the criteria to include molecular profiling studies in this analysis, since there are reports describing molecular platforms which are not listed here (e.g. https://journals.plos.org/plosone/article?id=10.1371/journal.pone.0224227). Lines 126-131 are an attempt to describe the methodology. Please put in a separate paragraph/section.
    4. Table 1 needs footnotes to explain abbreviations.
    5. What is the meaning of the column “% of targeted therapy”. Does this mean patients that ultimately received targeted therapy? If yes, please explain more accurately in that heading.

Minor:

  • Line 8: I think there is a typo mistake “ with the goal is…”. It may be rather with the goal of or which goal is. Please correct.
  • Line 65-73: This is a whole section of limitations, difficulties, etc. I think that this sort of information may be better placed under section 13 or 14 where this is discussed in the context of the current perspectives of molecular profiling. More concrete, lines 65 to 73 are not limited to adults but also children. Also
  • Lines 82-98 need to be more accurately referenced after each block of information provided.
  • Line 92: What do the authors want to mean by “stressing the need…”? I guess they want to highlight the fact that mutational analysis only may not be sufficient to identify other alterations such as CNV or structural variations which may only be got by means of WES. If the authors agree, a separate small section/table with a brief description of current available technologies (WES, RNAseq, etc) and pros/cons may be very useful for the reader to understand what are the limitations and scope of each technique and how/when to use them. They do an attempt on line 173 that may be expanded. Otherwise, the message the authors want to transmit here is diluted and not clear.
  • Line 105: What do the authors want to mean by (pilot). Much of them are not pilot anymore. In this same paragraph, I do not see much interest on commenting on the fact that the SM-PAEDS study is not yet reported, as it may be the case for the SPECTA from the EORTC for instance, or others that are still running. Please consider to remove or explain better the fact why to keep this. I think this may be better solved when the authors explain more accurately the methodology to get to these studies.
  • Line 115: “the feasibility and potential (…)”. What do they want to mean with “potential”? Potential interest, potential benefit, or simply potential as an adjective?
  • Line 144: I think that the discussion of when including these patients is larger than the two only time points the authors mention. It is debatable whether to include all R/R patients, or whether they consider HR also to these R/R or those who do not achieve meaningful responses after induction which is known to be a poor prognostic factor.
  • Line 331-339: The description in a row of three studies (ESMART, etc) is relatively superficial and does not capture the essence of these studies. I am not sure how of interest is to use these descriptions. I think better to remove. Also the last sentence (Line 336-338) is rather obvious and maybe superfluous.
  • Line 341-351: This is better under section 13/Discussion. Please consider my other comment on to group all these elements that are of debate under a single section which makes more identifiable the concepts that the authors want to highlight and debate.
  • Line 339: This definition of umbrella / basket studies is out of context. Please remove.
  • Table 1: Heading program name, consider also adding below “Sponsor” since the letters in italic refer to the sponsor.
  • Some genes are listed in italics while others in regular. Please homogenize.
  • Not all abbreviations listed in the abbreviations section appear in the text and not all abbreviations (other than titles of studies) are explained in the text. Please make sure all is explained and listed.
  • I guess the fact that the authors have opted for putting supplementary tables is because of editorial restrictions. But they have so far only used one table and one figure. I wonder whether 3 more tables or one table with three different sections in addition can be put as regular table so all is in the same document to facilitate accessing this information.
  • How the authors calculate the mean of 35 days in Line 271 if the information in S3 table is provided in means, ranges, average from each study?
  • Hereby I provide some references that the authors may consider to enrich some parts of the discussion or comment on additional topics that I find relevant for the manuscript:
    • The authors, when talking about the delayed effects of molecular therapy, may want to consider comment on the ACCELERATE initiative on this respect: https://onlinelibrary.wiley.com/doi/abs/10.1002/pbc.29047
    • An interesting issue is how the portfolio of cancer clinical trials in children has evolved over the last years with the progressive incorporation of molecular requisites to include patients. A recent report from the Spanish Group describe this change in the paradigm with a double proportion of studies in the second part of the study requiring molecular alterations to include patients. Maybe the authors can consider this topic to be discussed under section 7 or 10.
    • The authors may want to comment on the results of some paired therapeutic trials to those molecular platforms. The results of Arm D of ESMART are already published (https://pubmed.ncbi.nlm.nih.gov/33892407/)

Author Response

Comments Reviewer #1:

(line numbers refer to the version REVISIONS_CLEAN)

Major:                   

  • The authors need to explain what has been the methodology to identify and describe the precision medicine studies that they describe in section 4:
    1. Not all the studies in Figure 1 appear in Table 1 or in the text (lines 105-114). For instance, the PRISMA study only appears in Figure 1 but not in the other two parts. Please make sure there is a consistent list of studies in the three parts (text, figure and table). If not all studies have to appear in all three parts, then please explain the reasoning. – Thank you for pointing out these inconsistencies. We have made sure all 22 programs are mentioned in text (lines 120-127), the revised table 1 as well as the revised figure 2.
    2. Please, also explain how the size of the circumference has been estimated: Is it by samples included/samples analyzed/patients included? If it is patients included, please then add a column with patients included in Table 1 besides the columns of samples included/samples analyzed. – Thank you for the suggestion how to improve the clarity of the figure. We opted to add the number of samples included (not patients) to the circles instead of adapting the size to the number of samples included. We removed the orange color as well as the diagonal bar. We hope the simplified figure is more easy to understand.
    3. Please, describe what has been the search methodology to get to the published reports (e.g. timeframe of research [from 20xx to 20xx, date of last research performed, consort diagram, keywords, limits, etc; these are some examples, not all needed to be given but at least to be considered to be included).
      I particularly miss the pediatric study PROFILE (https://ascopubs.org/doi/full/10.1200/PO.20.00023); also, not clear to me what has been the criteria to include molecular profiling studies in this analysis, since there are reports describing molecular platforms which are not listed here (e.g. https://journals.plos.org/plosone/article?id=10.1371/journal.pone.0224227). Lines 126-131 are an attempt to describe the methodology. Please put in a separate paragraph/section. – Thank you for pointing this out. We added a methods section after the introduction and added the above mentioned studies to the text, table 1 and figure 2.
    4. Table 1 needs footnotes to explain abbreviations. – We added the abbreviations as suggested as footnotes to table 1. These were included in the submitted manuscript, however moved to abbreviations list only – editor’s choice?).
    5. What is the meaning of the column “% of targeted therapy”. Does this mean patients that ultimately received targeted therapy? If yes, please explain more accurately in that heading. – This is the percentage of patients receiving targeted therapy of all samples analyzed. We have added this explanation in more detail in the respective column heading.

Minor:

  • Line 8: I think there is a typo mistake “ with the goal is…”. It may be rather with the goal of or which goal is. Please correct. – Thank you, typo corrected to: “with the goal to improve…”(line 10).
  • Line 65-73: This is a whole section of limitations, difficulties, etc. I think that this sort of information may be better placed under section 13 or 14 where this is discussed in the context of the current perspectives of molecular profiling. More concrete, lines 65 to 73 are not limited to adults but also children. Lines 65-73 were removed and merged with the discussion in section 13d (lines 418-420).
  • Lines 82-98 need to be more accurately referenced after each block of information provided. – We have updated the references after each block in lines 82-98, as suggested.
  • Line 92: What do the authors want to mean by “stressing the need…”? I guess they want to highlight the fact that mutational analysis only may not be sufficient to identify other alterations such as CNV or structural variations which may only be got by means of WES. If the authors agree, a separate small section/table with a brief description of current available technologies (WES, RNAseq, etc) and pros/cons may be very useful for the reader to understand what are the limitations and scope of each technique and how/when to use them. They do an attempt on line 173 that may be expanded. Otherwise, the message the authors want to transmit here is diluted and not clear. – We drafted a figure to illustrate techniques used in the precision medicine programs including which type of event is detected by the specific method used with common examples in pediatric oncology (Figure 1). We hope this makes it more clearly how different techniques may be complementary indeed.
  • Line 105: What do the authors want to mean by (pilot). Much of them are not pilot anymore. In this same paragraph, I do not see much interest on commenting on the fact that the SM-PAEDS study is not yet reported, as it may be the case for the SPECTA from the EORTC for instance, or others that are still running. Please consider to remove or explain better the fact why to keep this. I think this may be better solved when the authors explain more accurately the methodology to get to these studies. – We have removed “pilot”. The SM-PAEDS study was added to text, table 1 and figure 2. The methods section has been added, as suggested in the major comments. We hope this is more clearly.
  • Line 115: “the feasibility and potential (…)”. What do they want to mean with “potential”? Potential interest, potential benefit, or simply potential as an adjective?  Changed to ‘opportunities” for readinig.
  • Line 144: I think that the discussion of when including these patients is larger than the two only time points the authors mention. It is debatable whether to include all R/R patients, or whether they consider HR also to these R/R or those who do not achieve meaningful responses after induction which is known to be a poor prognostic factor. – We hope by adding the subgroup of primary refractory patients explicitly to the section to underline the opportunities for these patients with a poor prognosis.
  • Line 331-339: The description in a row of three studies (ESMART, etc) is relatively superficial and does not capture the essence of these studies. I am not sure how of interest is to use these descriptions. I think better to remove. Also the last sentence (Line 336-338) is rather obvious and maybe superfluous. – We have removed the lines, as suggested.
  • Line 341-351: This is better under section 13/Discussion. Please consider my other comment on to group all these elements that are of debate under a single section which makes more identifiable the concepts that the authors want to highlight and debate. - Moved to discussion in section 13D, as suggested..
  • Line 339: This definition of umbrella / basket studies is out of context. Please remove. – Line removed.
  • Table 1: Heading program name, consider also adding below “Sponsor” since the letters in italic refer to the sponsor. – We added “Sponsor” to the heading of the first column in table 1.
  • Some genes are listed in italics while others in regular. Please homogenize. – All genes are listed in italics in the revised version.
  • Not all abbreviations listed in the abbreviations section appear in the text and not all abbreviations (other than titles of studies) are explained in the text. Please make sure all is explained and listed.- We verified all abbreviations are explained and listed. Some don’t appear in the text but in the table only (aCGH for example). It might be editor’s choice whether to keep these footnotes to table 1 or move to abbreviations section (as explained before)?
  • I guess the fact that the authors have opted for putting supplementary tables is because of editorial restrictions. But they have so far only used one table and one figure. I wonder whether 3 more tables or one table with three different sections in addition can be put as regular table so all is in the same document to facilitate accessing this information. - We added the information of the supplementary tables as separate colums to the main table (Table 1).
  • How the authors calculate the mean of 35 days in Line 271 if the information in S3 table is provided in means, ranges, average from each study? – We opted to exclude the mean of 35 days. Instead, we added the separate information to the main table, if available.
  • Hereby I provide some references that the authors may consider to enrich some parts of the discussion or comment on additional topics that I find relevant for the manuscript:
    • The authors, when talking about the delayed effects of molecular therapy, may want to consider comment on the ACCELERATE initiative on this respect: https://onlinelibrary.wiley.com/doi/abs/10.1002/pbc.29047 - We added this to section 13d.
    • An interesting issue is how the portfolio of cancer clinical trials in children has evolved over the last years with the progressive incorporation of molecular requisites to include patients. A recent report from the Spanish Group describe this change in the paradigm with a double proportion of studies in the second part of the study requiring molecular alterations to include patients. Maybe the authors can consider this topic to be discussed under section 7 or 10. – Although we feel this is an important topic to discuss, we decided not to include this subject since it’s beyond the scope of this manuscript.
    • The authors may want to comment on the results of some paired therapeutic trials to those molecular platforms. The results of Arm D of ESMART are already published (https://pubmed.ncbi.nlm.nih.gov/33892407/)  - Added results as suggested under section 13d.

Reviewer 2 Report

The paper is interesting and is a very complete overview on the subject. The following are minor comments:

1- line 97-99 a reference should be added

2-figure 1: it should be simplified and made more easy to understand, as it is nw it is not clear and immediate enough.

Author Response

The paper is interesting and is a very complete overview on the subject. The following are minor comments:

1- line 97-99 a reference should be added -  Thank you, we have added references as suggested,

2-figure 1: it should be simplified and made more easy to understand, as it is nw it is not clear and immediate enough. – Thank you for the suggestion how to improve the clarity of the figure. We opted to add the number of samples included (not patients) to the circles instead of adapting the size to the number of samples included. We removed the orange color as well as the diagonal bar. We hope the simplified figure is easier to understand.